# Organic Solar Cells Improved by Optically Resonant Silicon Nanoparticles

**DOI:** 10.3390/nano12213916

**Published:** 2022-11-06

**Authors:** Maria Sandzhieva, Darya Khmelevskaia, Dmitry Tatarinov, Lev Logunov, Kirill Samusev, Alexander Kuchmizhak, Sergey V. Makarov

**Affiliations:** 1School of Physics and Engineering, ITMO University, St. Petersburg 197101, Russia; 2Ioffe Institute, Russian Academy of Sciences, St. Petersburg 194021, Russia; 3Far Eastern Federal University, Vladivostok 690091, Russia; 4Institute of Automation and Control Processes, Far Eastern Branch, Russian Academy of Science, Vladivostok 690041, Russia; 5Harbin Engineering University, Harbin 150001, China; 6Qingdao Innovation and Development Center, Harbin Engineering University, Qingdao 266000, China

**Keywords:** organic solar cells, silicon nanoparticles, Mie resonances, laser ablation

## Abstract

Silicon nanophotonics has become a versatile platform for optics and optoelectronics. For example, strong light localization at the nanoscale and lack of parasitic losses in infrared and visible spectral ranges make resonant silicon nanoparticles a prospect for improvement in such rapidly developing fields as photovoltaics. Here, we employed optically resonant silicon nanoparticles produced by laser ablation for boosting the power conversion efficiency of organic solar cells. Namely, we created colloidal solutions of spherical nanoparticles with a range of diameters (80–240 nm) in different solvents. We tested how the nanoparticles’ position in the device, their concentration, silicon doping, and method of deposition affected the final device efficiency. The best conditions optimization resulted in an efficiency improvement from 6% up to 7.5%, which correlated with numerical simulations of nanoparticles’ optical properties. The developed low-cost approach paves the way toward highly efficient and stable solution-processable solar cells.

## 1. Introduction

Organic solar cells (OSCs) are a prospective class of devices owing to their solution fabrication techniques, high stability, and ability to be flexible, semitransparent, or even colorful, which is important for building-integrated and wearable photovoltaics [1,2,3,4]. Although the harvesting efficiency of OSCs is constantly increasing, there are still many issues that have to be addressed before commercialization [5]. To achieve highly efficient OSCs, one of the remaining issues is associated with the relatively low light absorption of organic photoactive layers. The film thickness is limited to 200 nm due to the low charge carriers’ mobility and short exciton diffusion lengths in OSCs (to minimize charge recombination losses). Thus, optical losses in the photoactive layer may reach ~40% of total losses [6,7]. In this way, the augmentation of light absorption of a photoactive layer of a solar cell (SC) at a fixed thickness is a promising approach to improve SC characteristics while preserving its other advantages. Many strategies have been developed for light management in OPV to decrease optical losses and enhance power conversion efficiency (PCE), including the design of more advanced donor/acceptor systems [8], as well as the development of new ternary or tandem SCs [9,10] and light trapping nanostructures [7,11]. In this regard, metallic nanostructures supporting surface plasmons polaritons in visible range incorporated in SCs can provide additional localization of the optical field on a deeply subwavelength scale, as well as enhance light absorption via an increase in light scattering [12,13]. As a result, the excited surface plasmons increase the absorption in a photo-active layer, thus enhancing light harvesting in a solar cell [14,15]. Incorporation of metal nanostructures has been applied to various PV technologies, such as dye-sensitized solar cells [16,17,18], silicon solar cells [19,20], organic photovoltaic cells [21,22,23], and perovskite solar cells [24]. In contrast to metallic NPs, all-dielectric nanophotonics based on silicon nanoparticles (Si NPs) with Mie resonances in visible and infrared ranges recently emerged as a powerful tool for various optical applications [25,26,27,28,29,30]. Owing to their low cost and chemical and temperature stability, Si NPs represent a viable and better alternative to noble metals for photovoltaic application. It should be noted that many types of ternary OPV have been developed, including hybrid organic-inorganic SCs that embedded Si NPs measuring approximately 2–5 nm [10] and 20–90 nm [31] in the active layer. However, the reported Si nanocrystals are too small to support Mie resonances in the visible range; the precise origin of PCE enhancement has been correlated with improved charge transport via better cascade energy-level alignment in the case of p-type Si NPs integrated into the bulk heterojunction (BHJ) layer.

Here, for the first time to our knowledge, optically resonant Si NPs were employed to improve OSC device performance, which boosted the final PCE from 6% (without Si NPs) up to 7.5% (with Si NPs). This approach paves the way for novel optimization strategies to improve organic photovoltaics by exploiting Mie-enhanced absorption in a photoactive layer by means of stable, chemically inert, low-cost, and sustainable Si NPs as compared to noble metal NPs.

## 2. Results

### 2.1. Samples

We fabricated Si NPs using the laser ablation method [32], which allowed for the creation of clean NPs in various solvents. To synthesize quasi-spherical Si NPs we used the laser ablation method in different solvents according to the procedure mentioned in reference [33]. As a source of highly intensive laser pulses generating NPs from Si targets, we used a commercial femtosecond laser system (Ti:Sa oscillator with a regenerative amplifier, Avesta Project Ltd., Moscow, Russia). The 40-fs laser pulses possessed 810 nm central wavelengths and a maximum pulse energy of 2.5 mJ at a 1 kHz repetition rate. The laser intensity was changed automatically using an acousto-optical modulator (R23080-3-LTD, Avesta Project Ltd., Moscow, Russia) and controlled by a power meter. Laser pulses were focused using a lens with a focal length of 5 cm (its focal beam diameter was approximately 150 µm) on a 1000-µm thick monocrystalline Si wafer covered by approximately 2 mm of solvent. Both p-type and n-type Si wafers were investigated. The Si NPs obtained from p-type wafers are herein referred to as ‘p-Si NPs’, and those obtained from n-type wafers are herein referred to as ‘n-Si NPs’. The concentration of Si NPs colloid was measured to be approximately 3 × 10^−4^ mol L^−1^ and might be optimized using a centrifugation procedure. Concentrations of synthesized samples were calculated via measurement of the optical absorbance of the solution in visible range (Shimadzu UV-3600, Shimadzu Ltd., Japan) and estimated using Mie theory. To prepare isopropanol-based Si NPs ink, we used a solvent exchange procedure via centrifugation. Namely, Si NPs in water were centrifuged at 10,000 rpm for 10 min and supernatant was removed. The precipitated Si NPs were dispersed in isopropanol and sonicated for 15 min using a high-power ultrasonic horn. Different concentrations of isopropanol colloidal inks were obtained by adding different amounts of solvent to precipitates of Si NPs. The obtained solution of Si NPs, shown in Figure 1a, exhibited a characteristic brown color, which indicated increased optical absorption in a short-wavelength range. The average diameter of NPs was approximately 150 nm according to our dynamic light scattering (DLS) measurements (Figure 1b). It is worth noting that direct laser ablation of Si NPs in isopropanol was much less efficient than in water.

In order to develop an efficient OSC, we employed the band structure design shown in Figure 1c; movements of electrons and holes are shown by arrows. Chemical and repeat unit structures of acceptor and donor polymer molecules used in the study are shown in Appendix A. Figure 1d shows an architectural schematic of the studied organic solar cells, representing a five-layer system, Ag(100 nm)/MoOx(8 nm)/ITIC:Ptb7-Th(120 nm)/ZnO(30 nm)/ITO(300 nm), deposited on a glass substrate. This architecture exhibits high stability and a reasonable efficiency level. In such a design, sunlight passes through ITO and ZnO to the photoactive absorbing layer. According to previous studies [14,24,34], to achieve the most optimum interaction of the incident light with resonant NPs and further its conversion to electricity, they are usually placed before the absorbing layer. Therefore, it was reasonable to consider two different positions of the layer with Si NPs: (i) above ITO or (ii) above ZnO layers, as shown schematically in Figure 1d. A typical cross-sectional SEM image of the obtained device with Si NP is shown in Figure 1e, which demonstrates the modified local morphology of the device.

### 2.2. Devices Fabrication and Characterization

The devices were fabricated on Indium tin oxide (ITO) patterned glass with an inverted configuration of ITO/ZnO/active layers/MoOx/Ag to investigate the photovoltaic performance of the Si NPs-modified organic solar cells. Patterned indium tin oxide (ITO) films, with a sheet resistance of 15 Ω/□ on the glass substrates, were cleaned in detergent, deionized water, acetone, and isopropanol using an ultrasonic bath, in sequence. The cleaned ITO films on glass were dried using nitrogen gas and then treated in an UV-ozone cleaner for 20 min. The ZnO precursor solution was prepared by dissolving 1 g of zinc acetate dihydrate (Zn(CH_3_COO)_2_*2H_2_O, Sigma-Aldrich, St. Louise, MO, USA, 99.9%) and 0.28 g of ethanolamine (NH_2_CH_2_CH_2_OH, Sigma-Aldrich, St. Louise, MO, USA, 99.5%) in 10 mL of 2-methoxyethanol (CH_3_OCH_2_CH_2_OH, Sigma-Aldrich, St. Louise, MO, USA, 99.8%). The ZnO precursor was spin-coated at 3000 rpm onto the ITO surface. After being annealed at 200 °C for 60 min in ambient conditions, the Si NPs colloidal inks were deposited onto the surface ZnO- or ITO-coated substrates using a spin-coating (1000 rpm, 60 s) or spray-coating technique (see more details in Appendix A). The bare and Si NPs-decorated substrates were transferred into a glove box. A solution of PTB7-Th (PBDTTT-EFT, Poly-[4,8-bis(5-(2-ethylhexyl)thiophen-2-yl)benzo [1,2-b;4,5-b’]-dithiophene-2,6-diyl-alt-(4-(2-ethylhexyl)-3-fluorothieno [3,4-b]thiophene-)-2-carboxylate-2–6-diyl)], PCE10, Ossila Ltd, Sheffield, UK ) and nonfullerene acceptor ITIC (3,9-bis(2-methylene-(3-(1,1-dicyanomethylene)-indanone)- 5,5,11,11-tetrakis(4-hexylphenyl)-dithieno [2,3-d:2′,3′-d′]-s-indaceno [1,2-b:5,6-b′]-dithiophene, 1-Material Inc., Quebec, Canada) in chlorobenzene with a concentration of 14 mg mL^−1^ (PTB7-Th:ITIC (1:1)) was spin-cast at 1000 rpm for 60 s. The photoactive layer films were vacuumed for 10 min. Next, a thin layer of MoOx film (≈8 nm) was evaporated, followed by Ag anode deposition (≈100 nm) via thermal evaporation. The active area of the devices was 21 mm^2^. The current density–voltage (J-V) characteristics were measured using a Keithley source measure unit (Tektronix, Beaverton, OR, USA). Solar cell performance was measured using an Air Mass 1.5 Global (AM 1.5 G) solar simulator (HAL-320, Asahi Spectra Inc., Torrance, CA, USA) with an irradiation intensity of 100 mW cm^−2^. The IPCE spectra for the inverted structure OSCs were measured using an IPCE measuring system (MAX-303, Asahi Spectra Inc., Torrance, CA, USA) with a CMS-100 monochromator.

Colloidal inks of Si NPs with an average size of approximately 150 nm and relatively broad size distribution (from 80 to 240 nm) (Figure 1a) were used in this work; these have polycrystalline structures with almost spherical shapes, according to previously reported data regarding the laser ablation approach to Si NPs synthesis [35].

Dielectric NPs’ position in the device structure is an important factor that affects the enhancement mechanisms of OPV devices. The distance between the photoactive layer and the dielectric resonant particles has a direct impact on the efficiency of the device. At a small distance, nonradiative electron-hole recombination might occur. Thus, the presence of an intermediate spacer between the dielectric Si NP and PAL may be necessary. However, it should be taken into account that the effect of local amplification has a near-field character. In this regard, two possible interfaces were chosen as possible surfaces for the Si NPs integration: the surface of ITO or zinc oxide ZnO, as shown in Figure 2. In the first case, an aqueous or isopropanol dispersion was directly spin-coated onto the bare ITO layer. In the second case, the spherical Si NPs were deposited on the ZnO layer using a spin-coating or spray-coating technique. The substrates were heated at 110 °C to remove residual solvent. The surface morphologies of the Si-decorated ZnO electron transport layers were investigated using optical microscopy.

Spin-coating deposition of Si NPs onto the ZnO layer showed a random but uniform dispersion of spherical Si NPs. Diluted Si NPs dispersions (C1) covered the surface uniformly without any agglomeration; however, highly concentrated solutions (C2) covered the surface non homogeneously, resulting in small agglomerations with large accumulations of nanoparticles (see Appendix Aa,b). The visualization of hundreds of nanoparticles in a dark-field microscope resembled a “starry sky”, where the colored spots corresponded to Si nanoparticles. The particle size distribution was in the range of 100–200 nm, which is in full agreement with the dynamic light scattering method (Figure 1b). Spray-coating from isopropanol colloidal dispersions led to coffee-ring shaped agglomerates on the ZnO surface (Appendix Ac).

The substrates were used in the fabrication of PTB7-Th:ITIC bulk heterojunction (BHJ) OSCs. A set of 32 devices was separated into four independent batches of production to study the influence of Si NPs light localization on photovoltaic parameters, as shown in Figure 2 and Appendix A. It should be noted that the incorporating Si NPs on the buffer ZnO layer did not worsen deposition of the blended photoactive layer, which means there was no deterioration of the interfacial morphology and wetting properties of ZnO. All OSCs (with and without Si NPs) showed a PCE higher than 6.5%, which corresponds to the standard state-of-the-art fabrication process for PTB7-Th:ITIC-based OSCs with a similar device area [36]. In the initial assessment, the champion reference device (Figure 2a) showed a power conversion efficiency (PCE) of 6.7%, which is one of the average results for PTB7-Th:ITIC-based OSCs. The devices with Si nanoparticles arrays in contact with ZnO showed significantly lower photovoltaic performance than the reference devices, with an average PCE of approximately 6%. In samples with nSiNPs_ZnO, open circuit voltages were 10% higher than the 800 mV open circuit of the reference devices. In contrast, the short circuit current density values were approximately 15% lower than that of the reference devices. This finding indicates that the Si NPs array in contact with the ZnO electron transport layer was not compatible with effective charge extraction. In contrast, both the Voc and Jsc of PTB7-Th:ITIC BHJ OSCs were slightly increased in p-Si_ZnO samples, in which random Si nanoparticles were sandwiched between electron conductive layers of ZnO and a photoactive layer. Direct contact of p-type Si NPs with the BHJ active layer seems to have been favorable for PCE enhancement. A clear increase in PCE from 6.0% (reference cell) to 7.5% (with pSi NPs on ZnO) was observed; it was based on increases in Jsc (+1%), Voc (+2%), and FF (+13%). A slight decrease in OSCs performance was observed in p-Si_ZnO type devices when higher concentrations of Si NPs (6 × 10^−4^ mol L^−1^) were used, as shown in Figure 3a–c. A higher concentration of Si NPs led to a slight worsening of morphology due to Si NPs agglomeration, resulting in decreases in all main parameters: Jcs, Voc, and FF (Figure 3d–f). Thus, the developed architecture with C1 concentration was close to the most optimal one. As for the spray-coated devices with p-Si NPs sandwiched between ZnO and BHJ layers, the general efficiency decreased compared to reference cells.

### 2.3. Discussion

To better understand the mechanism behind the enhancement of the devices’ PCE, we carried out numerical modeling of optical properties of a spherical Si NP placed inside a medium similar to that of the photoactive layer of our organic solar cells ITIC:Ptb7-Th (Figure 4a). A commercial software (CST Microwave Studio) was used to calculate the resonant responses of the Si NPs corresponding to those used in our experiment (with diameters from 80 to 200 nm) as well as absorption spectra for the ITIC:Ptb7-Th layer with and without Si NPs. The incident light was modeled as a plane wave in the 400–800 nm spectral range (Figure 4b). Standard open-boundary conditions provided calculations without reflections or disturbance of the near-field. The results of the modeling revealed several peaks in scattering spectra corresponding to different Mie modes, namely, electric and magnetic dipolar and quadrupole modes, which had characteristic near-field structure (see detailed description elsewhere [37,38,39]). More importantly, our numerical analysis revealed Si NPs’ optimal concentration, size, and space distribution, which led to an increase in active layer absorbance up to 9.4% around wavelength 600 nm.

Generally, the strongest absorption enhancement was observed in the 550–650 nm spectral range for all studied Si NPs. This was because Mie modes overlapped the wings of the ITIC:Ptb7-Th absorption band, as seen in Figure 4c. At maximum absorption, losses were too strong and suppressed efficient resonant properties, whereas in regions where losses on scattering and absorption were balanced (i.e., critical coupling conditions), the absorption efficiency was maximized. Similar critical coupling was widely employed for efficient optical heating of resonant nanoparticles [27,40]. Our numerical simulations and general discussion were also supported by experimental measurements of incident photon-to-current efficiency (IPCE) for the devices with and without Si NPs in Figure 4d. Indeed, the efficiency of photon-to-current conversion was higher around the 500–700 nm spectral range, which confirmed the crucial role of Mie resonances for the solar cells’ improvement. Additionally, a more detailed consideration of the device bands’ structure with p-doped Si NPs (Appendix A) revealed transport properties improvement providing an efficient pathway for the generated charge carriers, which is similar to that previously reported for doped-SiNPs for perovskite solar cells [41].

## 3. Conclusions

We have proposed using optically resonant silicon nanoparticles fabricated via laser ablation in liquid to boost the power conversion efficiency of organic solar cells. According to our numerical simulations, spherical Si NPs with diameters close to 150 nm demonstrated the best improvement in the devices; this is close to our experimental observations that nanoparticles with 140–180 nm diameters had a higher efficiency growth. The nanoparticles’ position in the device, their concentration, silicon doping, and method of NPs deposition were tested to check their effects on the final device efficiency. Parameters optimization resulted in an efficiency improvement from 6% up to 7.5%, which is better than previously reported improvements with plasmonic nanoparticles for similar organic solar cells materials (see comparison in Appendix A). This improvement was caused by matching the Mie resonances in Si NPs with certain spectral regions of the absorption band of the organic photoactive material, which correlated with numerical simulations of nanoparticles’ optical properties. This remarkable effect makes relatively narrow Mie resonances in Si NPs very promising for the improvement of solar cells based on organics with relatively narrow absorption bands, which provides some color effect and is suitable for building-integrated photovoltaic applications. We believe that further optimization of this technology can be achieved employing electrode designs [42] and new organic materials [43].

## Figures and Tables

**Figure 1 nanomaterials-12-03916-f001:**
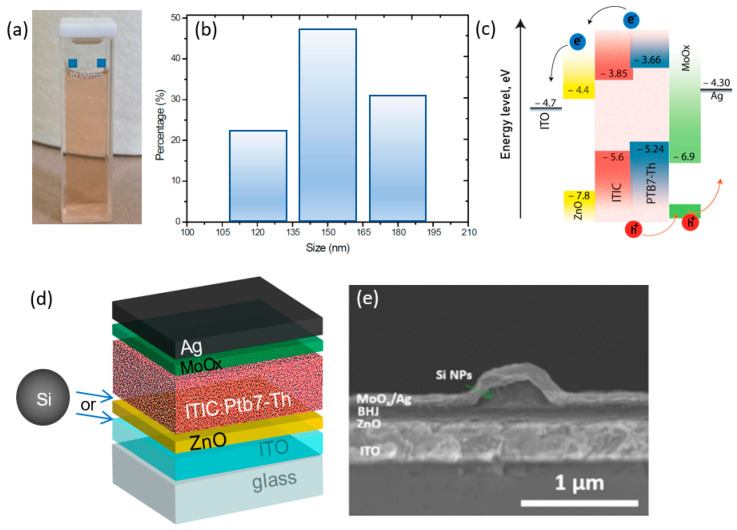
(**a**) Photo of a colloidal solution of Si NPs in isopropanol. (**b**) Size distribution of Si NPs in the solution. (**c**) Energy band diagram of the device. (**d**) Schematic illustration of the architecture of the studied organic solar cells. Arrows indicate the two positions of Si NPs deposition. (**e**) Cross-sectional SEM image of the obtained device with a single Si NP covered by BHJ/MoO_x_/Ag layers.

**Figure 2 nanomaterials-12-03916-f002:**
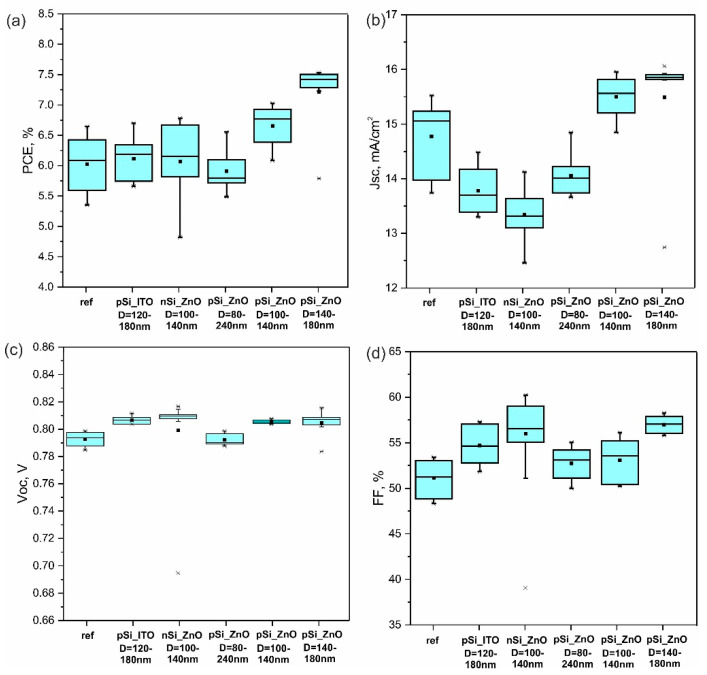
(**a**) PCE, (**b**) Jcs, (**c**) Voc, and (**d**) FF statistical data for photovoltaic performance of individual devices under 1 sun illumination for reference solar cells and for similar solar cells with different positions (on ITO or ZnO), diameters (D), and type of doping (nSi or pSi) of Si NPs. From left to right: reference cells without Si NPs; pSi NPs on ITO; nSi on ZnO; pSi NPs with D = 80–240 nm on ZnO deposited via spray-coating; pSi NPs with D = 100–140 nm on ZnO deposited via spin-coating with a concentration of 6 × 10^−4^ mol L^−1^; and pSi NPs with D = 140–180 nm on ZnO deposited via spin-coating with a concentration of 3 × 10^−4^ mol L^−1^.

**Figure 3 nanomaterials-12-03916-f003:**
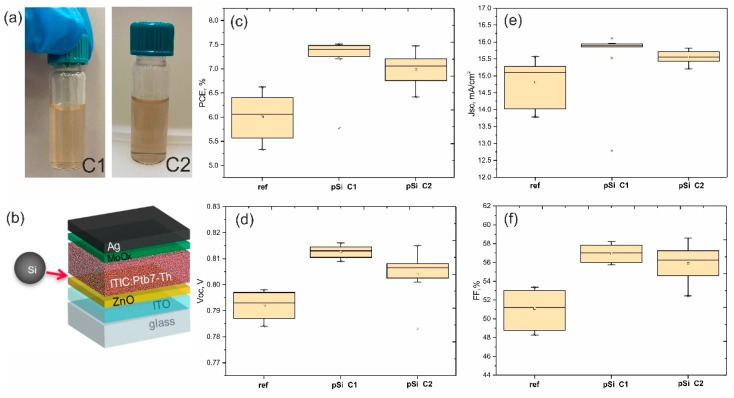
(**a**) Photographs of colloidal solutions of Si NPs in isopropanol with two concentrations: lower (C1) and higher (C2). (**b**) Schematic illustration of the architecture of the studied organic solar cell; an arrow indicates the position of the Si NPs layer. (**c**–**f**) PCE, Jcs, Voc, and FF statistical data for reference solar cells, and for similar solar cells with two different concentrations of Si NPs-C1 (3 × 10^−4^ mol L^−1^) and C2 (6 × 10^−4^ mol L^−1^).

**Figure 4 nanomaterials-12-03916-f004:**
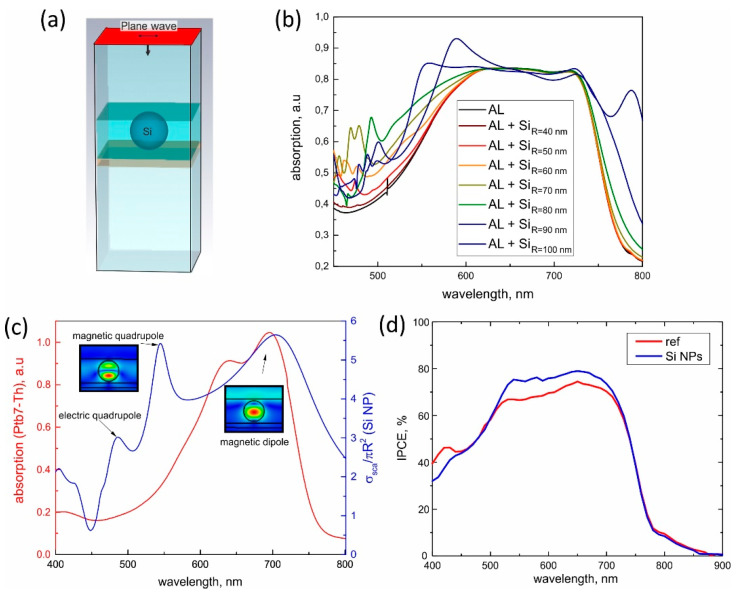
(**a**) A design for the numeral model. Red section corresponds to a port for the incident plane wave. (**b**) Numerically calculated spectra of absorption for the Ptb7-Th absorption layer (AL) without and with Si NP of different radii, from 40 nm to 100 nm. (**c**) Experimentally measured absorption spectrum of Ptb7-Th without Si NPs (red curve) and numerically calculated scattering efficiency of a single Si NP (with 150 nm diameter ) surrounded by Ptb7-Th (blue curve). Spectral peaks on the scattering spectrum correspond to different optical modes. (**d**) Experimentally measured incident photon-to-current efficiency (ICPE) for devices with and without Si NPs.

## Data Availability

The data presented in this study are available on request from the corresponding author.

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
