# Peer review of "Organic Solar Cells Improved by Optically Resonant Silicon Nanoparticles"

_nanomaterials, 2022, doi:10.3390/nano12213916_

Round 1

Reviewer 1 Report

The manuscript reported by Makarov and coauthors presents an original study about the insertion of optically resonant silicon nanoparticles produced by laser ablation technique into organic solar cells. In the work, authors study the potential location of the nanoparticles and their effect in the performance of the solar cell. The system is characterized by some techniques but, in my opinion, the characterization is not completely clear and more characterization would be welcome. Some aspects concluded are somewhat confusing and should be addresse by authors according to my comments below. On the other hand, authors should also improve the quality of many figures shown both in the manuscript and in the ESI file, because they possess low quality and many of them have been distorted (the wideness or height seems to be altered).

Comments:

-          The synthesis of the Si NPs performed by authors is very straight-forward, as detailed in the references. However, I wonder if additional trials have been done slightly modifying the conditions of the laser to better check if the efficiency of the process may be improved.

-          Moreover, I do not really understand the insertion of the NPs into the layers. In the manuscript (lines 84-87) it is said that the most appropriate situation of the NPs would be above ITO or above ZnO layers. However, in Figure 1e the NP seems to be placed between MoOx/Ag and BJH layers. Please provide any deeper explanation for this issue.

-          In the conclusions, authors state that “spherical Si NPs wit diameters close to 150 nm have demonstrated the best improvement of the devices.” However, in the final experiment conducted summarized in Figure 4, various Si NPs with different sizes are studied. Among those sizes, there is no rest for 150 nm, then I do not understand the previous conclusion.

-          I lack a general explanation about the Mie-theory in the supporting information.

-          There are some typo errors, such as “Zn(CH3COO)2*H2O” the asterisk in line 102.

-          The manuscript does not contain an “Experimental section” detailing the main methods and equipment employed in each case, although the latter are detailed during the results section.

-          In the Supporting Information file, please add the systematic names to the structures plotted in Figure S1. Moreover, the quality of those figures is not very good, I suggest authors to render them again with higher quality.

-          The plot shown in Figure S5 does not contain axes captions nor units. The quality is also too low.

Author Response

Reviewer 1:

General comment:

The manuscript reported by Makarov and coauthors presents an original study about the insertion of optically resonant silicon nanoparticles produced by laser ablation technique into organic solar cells. In the work, authors study the potential location of the nanoparticles and their effect in the performance of the solar cell. The system is characterized by some techniques but, in my opinion, the characterization is not completely clear and more characterization would be welcome. Some aspects concluded are somewhat confusing and should be addresse by authors according to my comments below. On the other hand, authors should also improve the quality of many figures shown both in the manuscript and in the ESI file, because they possess low quality and many of them have been distorted (the wideness or height seems to be altered).

Our Reply:

We are very thankful to the Reviewer for the careful and critical reading of your work. We agree with most of the comments, which encouraged us to improve the manuscript quality.

Comment:

-          The synthesis of the Si NPs performed by authors is very straight-forward, as detailed in the references. However, I wonder if additional trials have been done slightly modifying the conditions of the laser to better check if the efficiency of the process may be improved.

Our Reply: 

Generally, the interaction of a femtosecond laser pulse with bulk silicon in liquids can be performed in three different regimes: low-intensity, mid-intensity, and high-intensity. For the low-intensity mode, the so-called spallation mechanism dominates resulting in formation of relatively large particles because of pressure-induced detaching of a molten surface layer of silicon. The main disadvantage of this regime is its low rate of nanoparticles generation. For the high-intensity mode, the silicon surface undergoes so-called ‘phase explosion’ which is an ultrafast transition from solid state to supercritical fluid state, where small silicon clusters (diameters are much less than 100 nm) are formed mostly. Despite it yields a low number of nanoparticles, this regime is not suitable for Mie-resonant nanophotonics because of too small sizes. Thus, the most optimal regime is the mid-intensity one, which produces large enough sizes of nanoparticles and provides a reasonably high generation rate. On the other hand, we control the nanoparticles size just by centrifugation, which is almost equal to variation of laser intensity. We have added some short discussion to the manuscript on this point.

Regarding the variation of other parameters rather than intensity, in our additional experiments, we also tried ablation of silicon in two different organic solvents (chlorobenzene and isopropanol). The main result was that the generation of nanoparticles is much less efficient than that in water. The reason might be higher absorption of the laser radiation wavelength used by these solvents. Thus, when varying the conditions of laser ablation in water, it was found that in order to obtain enough silicon nanoparticles, it is necessary to provide a pulse energy of at least 1.1 mJ with a repetition rate of 1kHz. But in this case, the synthesis in chlorobenzene leads to rapid evaporation and blackening of the solution. The latter indicates the oxidation of an organic solvent takes place. Moreover, as a result of ablation, microparticles with a radius of more than 350 nm were obtained (Table S1). Ablation in isopropyl alcohol also leads to the formation of large silicon particles (more than a micron).

To summarize, we tried laser ablation in several liquids and optimized the regime of nanoparticles generation. The most optimal regime was revealed to be in water, and for their transfer to another solvent it is better to employ a two-stage method using centrifugation, decantation and subsequent dispersion by ultrasound. We have added some short discussion to the manuscript on this point.

Comment:

-          Moreover, I do not really understand the insertion of the NPs into the layers. In the manuscript (lines 84-87) it is said that the most appropriate situation of the NPs would be above ITO or above ZnO layers. However, in Figure 1e the NP seems to be placed between MoOx/Ag and BJH layers. Please provide any deeper explanation for this issue.

Our Reply: 

We are thankful for this question caused by some unclear description of Figure 1e. Indeed, we considered two different positions for Si NPs integration: (i) above ITO or (ii) above ZnO layers, as shown schematically in Figure 1d of the manuscript. And the cross-sectional SEM image of the obtained device with Si NP on Figure 1e depicts the case when NP is placed between ZnO and BHJ layers. Actually, in the picture we can see Si NP covered with an amorphous organic BHJ photoactive layer as some kind of “core - shell” structure In other words on the SEM image we do not see the Si NP directly, it is hidden under a sandwich of photoactive and MoOx/Ag layers and placed slightly in the depth of the picture, so we could see only a small part of spherical NPs under BHJ. This might lead to the wrong impression that Si NP is covered by Ag only. To avoid this misunderstanding, we have added an additional description for this picture.

Comment:

- In the conclusions, authors state that “spherical Si NPs wit diameters close to 150 nm have demonstrated the best improvement of the devices.” However, in the final experiment conducted summarized in Figure 4, various Si NPs with different sizes are studied. Among those sizes, there is no rest for 150 nm, then I do not understand the previous conclusion.

Our Reply: 

We agree that this part was written not clearly enough. Indeed, in the conclusion part, we stated the diameter size of Si NPs close to 150 nm to be appropriate and effective for PCE enhancement of organic solar cells, implying that this is the result of our calculations. On the other hand,  the average diameter of our Si NPs in colloidal inks is around 150 nm with relatively broad size distribution from 80 to 240 nm as mentioned in the beginning of section 2.2 and shown on Figure 1b of the manuscript. Additional centrifugation resulted in the narrowing of Si NPs size distribution to the range of 140-180 nm, being close to the most optimal condition. In order to clarify this part, we have rewritten our relevant statements in the conclusion.

Comment:

-          I lack a general explanation about the Mie-theory in the supporting information.

Our Reply: 

We agree that more descriptions for Mie modes and numerical simulations are important for better understanding. Therefore, we have added more explanations and references to the Discussion part.

Comment:

-          There are some typo errors, such as “Zn(CH3COO)2*H2O” the asterisk in line 102.

Our Reply: 

We agree with the comment and we are thankful for this notice. The misprint has been corrected in the main text of the manuscript to Zn(CH3COO)2*2H2O.

Comment:

-          The manuscript does not contain an “Experimental section” detailing the main methods and equipment employed in each case, although the latter are detailed during the results section.

Our Reply:

We agree that the “Experimental section” is very popular in many journals, separating the main results from some additional protocols and technical descriptions. However, we followed recommendations of the Nanomaterials and some examples of recent works in this journal. Nevertheless, we believe that in our work we managed to naturally merge the results with methods of experiments and calculations, while our Supplementary materials contain some additional less important information.

Comment:

-          In the Supporting Information file, please add the systematic names to the structures plotted in Figure S1. Moreover, the quality of those figures is not very good, I suggest authors to render them again with higher quality.

Our Reply: 

We have added the systematic names to the structures of photoactive materials used in this study and plotted in Figure S1. We also have redrawn the chemical structures of the materials to improve the quality of the image.

Comment:

-          The plot shown in Figure S5 does not contain axes captions nor units. The quality is also too low.

Our Reply:

We are thankful to the Reviewer for this useful notice. We have added axes description and improved the figure quality.

Reviewer 2 Report

The authors fabricated silicon nanoparticles by a laser ablation method and employed them in an organic solar cell structure for boosting power conversion efficiency. The paper is overall well-written, with both simplified simulations and experimental results.

Please see the comments below:

(1) It's hard to tell the shape of the silicon nanoparticles, especially from Fig 1e, it doesn't look spherical, please clarify this.

(2) I would suggest the authors add a table to compare the results with other previous papers using similar configurations and similar methods (i.e., metallic, or dielectric nanoparticles).

Author Response

Reviewer 2:

General comment:

The authors fabricated silicon nanoparticles by a laser ablation method and employed them in an organic solar cell structure for boosting power conversion efficiency. The paper is overall well-written, with both simplified simulations and experimental results.

Our Reply: 

We are very thankful for the positive evaluation of our work!

Comment:

(1) It's hard to tell the shape of the silicon nanoparticles, especially from Fig 1e, it doesn't look spherical, please clarify this.

Our Reply:

Yes, the Reviewer is absolutely right that laser ablation in liquid might give some non-spherical nanoparticles, because of their fast solidification process in a liquid environment. However, in many papers and in the provided reference [25], SEM images of ablated Si NPs reveal their quasi-spherical shape with deviation from the sphericity up to 10% (radius relative changes). On the other hand, in the papers on Mie resonances in Si NPs (see e.g. [Nanoscale 8 (18), 9721-9726 (2016)]), it was confirmed that slight ellipticity does not shift spectral position of the first two Mie renanonses, which give the highest contribution to PCE improvement in our case. We have rewritten the part related to Si NPs formation mentioning that they have quasi-spherical shape.

Comment:

(2) I would suggest the authors add a table to compare the results with other previous papers using similar configurations and similar methods (i.e., metallic, or dielectric nanoparticles).

Our Reply:

We thank the reviewer for highlighting this point and for a useful suggestion. Various metal nanoparticles have been used for PCE enhancement in OPV in different stacking layers. Recently, Au nanoparticles have been embedded in the ZnO electron transport layer of OSC using the same photoactive layer [Usmani, et al “Inverted PTB7-Th:PC71BM organic solar cells with 11.8% PCE via incorporation of gold nanoparticles in ZnO electron transport layer” Solar Energy 214, 220-230 (2021)]. The reported 24% enhancement in PCE was attributed to plasmonic effect of Au nanoparticles, which benefits light absorption in the active layer resulting in 15% photocurrent improvement and FF increase on 5%.  We have added Table S1 to address the Reviewer’s comment.

Reviewer 3 Report

In this paper, the authors report the use of high-refractive-index Si resonant NPs that apparently have positive effects on the efficiency of organic solar cells.

But the work is not convincing and the information is inconsistent. The introduction is too vague, and the degree of novelty is not clear because it is not clear from the paper how these nanoparticles are more special than those manufactured by other groups using the same method. The introduction does not specify the state of the art regarding the effect of Si Nps in OPVs as a function of their doping.

The authors claim that the size of the NPs is in the range of 84 - 240 nm , and are situated at the n-type interface, yet in fig 1e) a Si NP is shown with at least 400 nm size at the p-type interface, on top of the BHJ.

The entire paper analyzes the Si Nps, yet "The surface morphology of the Si-decorated ZnO ETLs were investigated using optical microscopy"? How can the dispersion of NPs on a surface be visible with optical microscopy? 

It is not clear how were they separated and dispersed to fabricate solar cells with different diameter NPs

Author Response

Reviewer 3:

General comment:

In this paper, the authors report the use of high-refractive-index Si resonant NPs that apparently have positive effects on the efficiency of organic solar cells.

Our Reply: 

We are thankful for the careful reading of our work, and we have divided his/her opinion onto several separated comments for more convenient reading of our replies.

Comment:

(1) But the work is not convincing and the information is inconsistent. The introduction is too vague, and the degree of novelty is not clear because it is not clear from the paper how these nanoparticles are more special than those manufactured by other groups using the same method.  The introduction does not specify the state of the art regarding the effect of Si Nps in OPVs as a function of their doping. 

Our Reply: 

First of all, we would like to stress here that Mie-resonant Si NPs were applied for the first time for OPV, which is the key novelty of our work. However, we agree that providing more context on the mentioned Si NPs in OPVs would improve the manuscript quality. Thus, we have rewritten the introduction more carefully and added the information on two papers about hybrid ternary SCs  that used Si nanoparticles inside of the BHJ active layer, which we believe now improves the clarity of our results

It should be also emphasized that the reported before Si NPs had diameters around 2-5 nm [Zhao, et al “Silicon-nanocrystal-incorporated ternary hybrid solar cells, Nano Energy, 26, 305-312 (2016)] and 20-90 nm [Hemaprabha, et al. “Doped silicon nanoparticles for enhanced charge transportation in organic- inorganic hybrid solar cells” Solar Energy, 173, 744-751 (2018)]. These NPs are too small to support any Mie resonances in the visible range. Thus, the origin of PCE improving up to 21% has been correlated with improved charge transport in these studies. It is likely that better cascade energy-level alignment in the case of p-type Si NPs integrated in the BHJ layer facilitates charge separation compared to reference and cells with n-type Si NPs. 

In our work, we used resonant Si NPs with diameters in the range of 80-240 nm for boosting the OSC efficiency for the first time. Such Si NPs support optical Mie-resonances, strong light scattering in the visible wavelength range and optical enhancement in near-field, that allowed us to observe the PCE enhancement up to 25% via improved light absorption in OSCs. Our theoretical simulations and general discussion is also supported by experimental measurements of Incident-Photon-to-Current Efficiency (IPCE) for the devices with and without p-type Si NPs in Figure 4d of the Manuscript. Indeed, the efficiency of photons-to-current conversion is higher in the spectral range around 500-700 nm, which confirms the crucial role of Mie resonances for the solar cells improvement. Additionally, according to the more detailed consideration of the device bands structure with p-doped Si NPs reveals transport properties improvement similar to that reported before for doped-SiNPs for perovskite solar cells.

Comment:

(2) The authors claim that the size of the NPs is in the range of 84 - 240 nm , and are situated at the n-type interface, yet in Fig 1e) a Si NP is shown with at least 400 nm size at the p-type interface, on top of the BHJ.

Our Reply: 

Cross sectional SEM demonstrates the example of Si NP  incorporation in the structure of organic solar cell with inverted architecture ITO/ZnO/pSiNPs/BHJ/MoOx/Ag where p-doped Si NPs are deposited on top of ZnO layer (n-type interface). The whole stack of 4 functional layers without Si NPs can be represented as ZnO/BHJ/MoOx/Ag with total thickness around 260 nm (without ITO layer). In the center of the image (Figure 1e)  the modified stack of OSC with integrated Si NP with diameter around 150 nm is represented, so we could see how the organic photoactive layer wraps around the NP due to its amorphous nature and solution deposition of BHJ active layer above Si NPs leading to spherical type heterogeneity.  In other words, in the SEM image we do not see the Si NP directly, it is hidden under  photoactive and MoOx/Ag layers and placed slightly in the depth of the picture. Indeed, the overall thickness in the place of integration of Si NPs would be around 410 nm (260 nm +150 nm). To avoid this misunderstanding, we have added an additional description for this picture.

Comment:

(3) The entire paper analyzes the Si Nps, yet "The surface morphology of the Si-decorated ZnO ETLs were investigated using optical microscopy"? How can the dispersion of NPs on a surface be visible with optical microscopy? 

Our Reply: 

Spherical silicon nanoparticles ranging in size from 100 nm to several hundred nanometers exhibit strong Mie resonances in the visible range. In an optical microscope, such silicon nanoparticles can be seen in the dark-field mode due to strong resonant light scattering on these particles. As a result of light scattering in a dark-field microscope, the nanoparticles look like bright multicolored spots from purple to red (Figure S3). This method is quite standard for Si nanoparticles investigation [Kuznetsov, et al "Magnetic light." Scientific reports 2, no. 1 (2012): 1-6. ;  Evlyukhin, et al "Demonstration of magnetic dipole resonances of dielectric nanospheres in the visible region." Nano letters 12, no. 7 (2012): 3749-3755.].

Comment:

(4) It is not clear how were they separated and dispersed to fabricate solar cells with different diameter NPs

Our Reply: 

We have fabricated Si NPs in water solution by laser ablation method. To prepare isopropanol (IPA) based Si NPs ink, we used a solvent exchange procedure by centrifugation method as mentioned in section (2.1.) of manuscript. In more details: the obtained Si NPs in water were centrifuged at 10,000 rpm for 10 minutes and supernatant was removed. The precipitation with Si particles were dispersed in IPA and sonicated for 15 min using high power ultrasonic horn. Different concentrations in IPA colloidal inks were obtained by adding different amounts of solvent to precipitates of Si NPs. In order to clarify this part, we have rewritten our relevant descriptions.

Round 2

Reviewer 3 Report

Thank you to the authors for their concrete and reasoned answers to my observations.